# Deciding Alone or with Others: Employment Anxiety and Social Distance Predict Intuitiveness in Career Decision Making

**DOI:** 10.3390/ijerph20021484

**Published:** 2023-01-13

**Authors:** Xiaoli Shu, Jun Peng, Guilin Wang

**Affiliations:** 1School of Education Science, Hanshan Normal University, Chaozhou 521041, China; 2School of Education, Research Institute of Macau Education Development, City University of Macau, Macau 999078, China

**Keywords:** employment anxiety, social distance, intuitive career decision, intuitiveness

## Abstract

Intuitive career decisions can influence people’s career choices and subsequent job competencies, which are related to their development and happiness. There is evidence that both anxiety and social distance influence intuitive career decisions individually, but it is unclear how employment anxiety and social distance influence intuitive career decisions individually and how they interact to influence intuitive career decisions. Drawing on the cognitive–emotional dual-system model, in this study, 298 college students and 386 senior job-seeking students were tested through behavioral experiments and questionnaires, respectively. The results showed that employment anxious individuals have a higher intuitive level in career decision making, and they also have a higher intuitive level when making career decisions for others at a far social distance. In addition, employment anxiety and social distance interact to influence the intuitiveness of career decision making. When making career decisions for themselves and those who are close to them, the increase in employment anxiety will increase the intuitive level. Therefore, in a non-anxious situation, you can make career decisions on your own or get help from someone close to you, but in anxious situations, you can turn to others who are at a far social distance to help make decisions.

## 1. Introduction

Career decision making is an individual’s choice about the career he or she is going to pursue [1,2] and is one of the important decisions that people must face in their lives, which is related to their development and well-being and may have indirect social consequences for society [3,4]. In recent years, the factors influencing career decisions have received extensive scholarly attention [5,6,7], but these studies have analyzed the direct influences on career decisions in a piecemeal manner, making it difficult to provide an adequate theoretical basis for promoting effective decision making.

According to the cognitive–emotional dual-system model, there are two interactive information processing systems in decision making, namely, the cognitive system and the emotional system [8]. The cognitive system requires conscious thought, which mainly relies on logical rules and deliberate and systematic reasoning to make decisions, with low levels of intuition. The emotional system requires less conscious participation, mainly relying on experience and the surface perception of things to make decisions, which are somewhat intuitive [8,9]. The cognitive and emotional systems play different roles in the career decision-making process, as does their level of intuition [2,10]. According to Harren’s description, the basis for making intuitive career decisions often comprises “fantasy, attention to present feelings, and an emotional self-awareness” rather than “anticipation of the future, information-seeking behavior, or logical weighing of actors”. A decision to pursue a course of action is “reached relatively quickly”, and it is “less likely to result in effective decision making” [2].

When cognition is at play, people conduct rational, organized, and adequate analysis based on a search for relevant information about themselves and their professional environment and then make effective decisions with low intuitiveness [2,11]. People are able to form more objective and accurate self-concepts in the career decision-making process, better understand their occupational environment, make the best choice at that time, and later become more competent in their job [2]. However, when emotions are at play, instead of focusing on information searching and rational trade-offs between information, people make quick decisions based on their current emotions or emotional self-awareness, and these decisions are very intuitive [2,6,12]. Individuals use only limited information and emotion to understand their own occupational situations in the career decision process, with little information related to themselves and their occupational environment, so the decision is highly intuitive and may affect their job competence after work [2,11].

It is seen that exploring the factors influencing the intuitiveness of career decisions and the mechanisms occurring from both cognitive and emotional aspects can better explain how individuals make more effective career decisions [11]. This can be used to better understand individuals’ career choice behaviors and promote their career development [10,13].

### 1.1. Employment Anxiety and Intuitive Career Decision Making

One argument about the model of emotions in decision making was advanced by Loewenstein et al. (2001), who argued that individual decision-making behavior is influenced by anticipated emotions and immediate emotions [14]. Anxiety, which is common among many people, can be divided into state anxiety and trait anxiety. State anxiety is a short and immediate emotion that is a physiological activation and conscious fear, worry, or tension in a specific situation. Trait anxiety is a stable anticipated tendency regarding emotions and reactions [15], which both influence decision making [12]. Employment anxiety is “a restless emotional experience caused by college students’ bad cognition of employment goals, processes and results” [16], and is conceptualized as “a specific type of state anxiety evoked by employment concerns” [17]. With the uncertainty created by COVID-19, people experience increasing levels of employment anxiety during their career transition [18].

Anxiety affects decision making by influencing an individual’s cognitive processing task. First, anxiety affects the individual’s perception of information changes, and people with high anxiety see fewer changes [19]. Second, anxiety also has a direct impact on memory, and individuals with higher levels of state anxiety perform poorly in prospective memory tests [20]. Third, anxiety also affects the reasoning process. In integrating current information into original knowledge and relational integration reasoning, the performance of high-anxiety individuals is worse than that of low-anxiety individuals [21]. That is, people who are anxious have weakened cognitive abilities in perception, memory, and reasoning [19,20,21], which affects the collection and processing of information, increasing the intuitiveness of decision makers’ decisions [12].

In career decision making, reducing negative emotions as much as possible is one of the effective choice goals [22]. Some scholars have also found that individuals with high anxiety prefer low-return choices, and they will seek consistency in decision making, which will affect their independent decisions [23]. In addition, anxious individuals pay more attention to negative choices, and usually interpret information negatively. They tend to avoid potential negative results under vague situations even at the cost of missing out on opportunities. They usually cannot obtain comprehensive evaluation information [24]. In other words, they make decisions with less information, make fewer correct choices, and make more intuitive decisions [25].

Some scholars have also found that there is a positive correlation between career indecision and anxiety in career decision making [26]. Anxious people have low levels of confidence in making career decisions. They have more information about the treatment of career attributes in the process of career decision making, which takes a long time to understand [27]. More memory resources are required as state anxiety levels rise, and there is a corresponding decline in the amount of cognitive resources available for making decisions, which lowers an individual’s cognitive performance [20]. In such cases, individuals tend to collect less information to reduce their uncertainty before deciding [25]. As a result, they cannot fully consider all possibilities and their decisions are highly intuitive and less accurate [28]. Therefore, in the career decision-making process, it is believed that anxious individuals make more intuitive decisions. Based on the above theoretical derivation, we hypothesize the following:

**H1.** *Employment anxiety affects intuitiveness in career decisions, and individuals with high employment anxiety are more intuitive*.

### 1.2. Social Distance and Intuitive Career Decision Making

People in the information age usually make decisions for themselves or help others [29]. Studies have found that there are differences between making decisions for themselves and others [7,30,31,32]. Social distance, defined as “the affective closeness between themselves and others” [33], is a social situational cognitive variable that reflects psychological distance [34,35]. This topic has received much attention in the field of decision making.

It has been discovered that the social distance between the decision maker and the decisionee affects decision making [36,37,38,39] and we predict that social distance also influences career decision making. According to the Person–Job Fit theory, individuals need to know information about job seekers’ personality traits and career information and match them well before they can make a rational career decision, rather than an intuitive one [11,40]. Individuals who are close to each other have more interpersonal similarities, interact more, and know each other better [41]. This makes it possible for decision makers to collect more comprehensive information about job seekers, better match personality traits and career information, complete person–job matching, and make less intuitive career decisions without being influenced by other factors such as emotions [2,11]. Mutual recognition and understanding between individuals who are socially distant is low [33]. This results in decision makers not knowing job seekers well enough, and it will be difficult to complete person–job matching because of limited information, and they will make highly intuitive career decisions [11].

Furthermore, when making decisions for individuals with close social distance, people pay more attention to concrete, contextualized information, whereas when making decisions for individuals with far social distance, they pay more attention to abstract, decontextualized information [42]. Concrete, contextualized information helps decision makers match person–job information in the career decision-making process, but abstract, decontextualized information does not [40]. That is, it is more difficult for decision makers to complete the person–job match for individuals with far social distance than for individuals with close social distance. As a result, we believe that decision makers making career decisions for individuals with far social distance will be more intuitive. Therefore, we hypothesize the following:

**H2.** *Social distance affects the intuitive strategy of career decision making, and it is more intuitive for individuals who are at a far social distance to make career decisions*.

### 1.3. The Interaction of Employment Anxiety and Social Distance on Intuitive Career Decision Making

The existing research results on the influence of social distance on decision making are not uniform. Some scholars have found that the farther the social distance is, the more impulsive the decisions that are made [37,43]; however, others have found that with an increase in social distance, people become more rational, avoiding adverse consequences and seeking ideal results [33]. The inconsistency of these results shows that there are other factors influencing the interaction between social distance and decision making. According to the cognitive–emotional dual-system model [14], emotion is most likely to be one influencing factor between social distance and decision making. Considering the literature related to social distance, most of the studies focus on the cognitive processes of individuals, and most of them analyze the influence of social distance on decision making according to construal level theory. Loewenstein et al. put forward the idea of adding emotion channels into a decision-making model [14], and emotions have been gradually brought into research on the influencing mechanism of social distance. Previous research has investigated the effects of anxiety on decision making as well as the effects of social distance on decision making, but there is little research on the effects of anxiety and social distance on decision making [18], and there is almost no research on the effects of anxiety and social distance on career decision making.

From the emotional channel point of view, individuals can understand their emotional arousal more accurately than others can, but it is difficult to accurately perceive others’ emotional arousal [44]. The greater the social distance is, the more difficult it is to accurately understand others’ emotions, and an increase in social distance will reduce the emotional arousal of the decision maker regarding decision-making events [45]. Therefore, in anxiety-inducing situations, when individuals make decisions for themselves or others close to them, they are more likely to be aroused. However, when making decisions for others far from them, their anxiety is seldom aroused [44]. Williams et al. also found that when making decisions for others, the emotional experience of decision makers will not change much, regardless of whether it is a positive or negative event [45]. Fernandez-Duque and Wifall believed that deciding for themselves tends to start the emotional system channel and making decisions for others tends to start the cognitive analysis system channel [46]. In the decision-making process, the decisions one makes for oneself are more easily influenced by emotions than the decisions made for others. When individuals make career decisions for themselves and close to others during anxiety arousal or under the influence of anxiety, they are unable to consider the full range of possibilities for personal and career information, and they will make career decisions rashly and without strategizing [47], so their career decision making is more intuitive.

According to construal level theory, when individuals make career decisions for individuals far from them, they pay more attention to relatively coherent, abstract, and superordinate information, and this information is less influenced by their anxiety [35]. Moreover, the level of anxiety arousal is small at this time, and the anxiety that individuals feel is also small. Therefore, when making decisions for strangers, anxiety has little impact on career decisions. When individuals make career decisions for people close to them, they are more focused on detailed, specific information and personal preferences, and this information is more susceptible to anxiety [32]. Moreover, at this time, anxiety is more likely to be aroused and individuals are more anxious [44]. Therefore, decision makers assist those close to them in making decisions, anxiety has a significant impact on career decisions, and decisions are more intuitive. Guo et al. found that social distance moderated the process of uncertain decision making [38], and this study provided indirect evidence for the moderating of social distance. Based on the above analysis, we propose:

**H3.** *Anxiety and social distance interact to influence intuitive career decision making. That is, social distance moderates the relationship between anxiety and intuitive career decision making. When social distance is close, anxiety has a greater influence on the intuitive level of career decision making. However, the influence of anxiety on the intuitive level will be weaker when social distance is far*.

### 1.4. The Present Study

The purpose of the present study was to explore how anxiety and social distance influence career decision making and how they interact. In Study 1, starting with inducing anxiety, we used behavioral experiments to investigate the influence of anxiety and social distance on career decision making and then explored the interactive influence of anxiety and social distance on career decision making to answer the question of which situation needs the self or others to help make a decision. In Study 2, through the questionnaire survey of senior students who are looking for jobs, the moderating role of social distance in college students’ career decision making is further verified.

## 2. Study 1: An Experimental Study on the Influence of Anxiety and Social Distance on Intuitive Career Decision Making

### 2.1. Participants

G*Power3.1 was used to calculate the required sample size. Variance analysis was used as the statistical method, the effect size was set to 0.25, α was 0.05, the number of groups was set to 6 [48], and the total number of samples was calculated as 251. Considering the wastage rate of the samples, 320 college students from a university in China were randomly selected to take part in the experiment, and they were randomly assigned to either the high employment anxiety group or the low employment anxiety group. A total of 298 valid questionnaires were obtained, with a recovery rate of 93%. Among them, the high employment anxiety group had 182 participants (79 males and 103 females) who were in anxiety-inducing situations during the experiment, while the low employment anxiety group had 116 participants (52 males and 64 females) who were not. Before the experiment, the mean anxiety level for the high employment anxiety group was 2.13 (*SD* = 0.45) and 2.11 (*SD* = 0.40) for the low employment anxiety group, with no significant difference between the two groups (*t* = 0.40, *p* = 0.69, *d* = 0.02). The average age of the participants was 20.04 (*SD* = 1.59).

The local human ethics committee approved the research project. All the recruited participants gave informed consent in written form. The investigation was conducted from June 2021 to December 2021.

### 2.2. Procedure

First, employment-related anxiety was induced. Participants were randomly divided into two groups: the situation with induced anxiety and the situation with no induced anxiety. Second, the state anxiety scale was completed to test whether there was a difference in employment anxiety levels between the two groups to judge whether employment anxiety induction was successful. Third, the IOS scale was presented, and participants were asked to indicate on a social distance graph the distance between themselves and the decision maker. Finally, the question about intuitive career decision-making was answered. After the experiment, they received the reward of RMB 10 (approximately USD 1.50).

### 2.3. Materials and Measures

According to the translation/back translation process, we created Chinese versions of the State Anxiety Inventory and IOS Inventory. First, we asked an English teacher to translate into Chinese, then another English teacher to translate the translated Chinese back into English, and so on until the Chinese version was complete. Employment anxiety-inducing situations and intuitive career decision-making situations were consistent with real situations. We compiled the possible employment anxiety triggering situations and intuitive career decision-making situations through interviews with college students and career guidance teachers, and then asked college graduates to select the real situations that best matched.

#### 2.3.1. Employment Anxiety-Inducing Situation

Referring to the paradigm of social stress exposure adopted by Chajut and Algom [49], this study used the method of reading severe news related to employment to induce individual employment anxiety. The specific situation of employment-related induced anxiety was set as follows (high employment anxiety group): According to relevant surveys, in 2020, many internet-based and entity enterprises frequently closed down. JD, Tencent, Alibaba, and other enterprises frequently laid-off employees. Many enterprises internally spread the news that they had stopped recruitment efforts. According to statistics, the number of college graduates in China will reach 9.09 million in 2021, a record high. However, the number of new jobs available for college graduates will decrease by 49% year on year, and the unemployment rate will become more severe over the next few years. The group without employment anxiety induced (control group) was presented with a paragraph describing psychological concepts, with roughly the same number of words.

#### 2.3.2. State Anxiety Scale

Employment anxiety was measured using the State Anxiety Inventory developed by Spielberger et al. [50]. A scale of 20 items was used to measure state anxiety on a 4-point Likert scale ranging from 1 (almost never) to 5 (almost always), in which 10 points were the reverse scores. Participants were asked to choose the most realistic answer according to their perceived job selection situation upon graduation. The inventory contains questions such as “I am worried now, but I feel that my fears are more severe than what I may face in reality”. The higher the score, the more serious the anxiety. The Cronbach’s α coefficient in our sample was 0.91.

#### 2.3.3. The IOS Scale

The Inclusion of Other in the Self (IOS) scale developed by Aron et al. was used to measure social distance [51,52]. There are 7 pairs of circles with different overlapping parts. The overlapping part represents the degree of intimacy between oneself and decision-making others, and a higher number of overlapping parts represents a closer relationship. According to the research of Wu et al. [52], the degree of intimacy was divided into 3 levels, and a 3-point score was adopted, that is, a far relationship, medium relationship, and close relationship.

#### 2.3.4. Intuitive Career Decision-Making Situation and Operation Methods

According to Scott and Bruce [53], they argued that intuitive decision making does not pay attention to the collection and analysis of information and alternative information; they involve simply deciding in a short time based on the feeling or temporary impulse and premonition at that time. On this basis, this study created the following career decision-making situation.

There is a job opportunity. The individual briefly browses the relevant information and feels okay about the decision. The probability that he or she will immediately choose this job is presented.

The operation method here draws on the behavior difference method of Yang and Li to obtain continuous data [54]. Choices closer to the 0% end of the scale indicate a low level of intuitive career decisions and are scored as 1. Choices closer to the 100% end of the scale indicate a high level of intuitive career decisions and are scored as 5. Choices closer to the 50% end of the scale indicate an intermediate intuitive level and are scored as 3. As shown in Figure 1, a higher number means a higher intuitive level.

### 2.4. Statistical Analyses

Data were processed and analyzed mainly by the SPSS 23.0 Windows and Process 3.5 macro program. First, an independent sample *t*-test was used to analyze whether the anxiety induction was effective. Second, the descriptive statistics of variables and Pearson correlations were tabulated and tested. Third, a multivariate ANOVA was used to test the interaction between anxiety and social distance on the intuitiveness of career decisions, using a 2 (employment anxiety: high, low) × 3 (social distance: intimate, moderate, unfamiliar) experimental design, in which both employment anxiety and social distance were independent variables, variables were designed between subjects, and the dependent variable was the intuitive level of career decision making.

### 2.5. Results

#### 2.5.1. Induction Effect of Employment Anxiety

The average score of state anxiety in the high employment anxiety group was 2.61 (*SD* = 0.40) and that of the low employment anxiety group was 2.15 (*SD* = 0.42). The independent sample t-test results showed that the state anxiety level of the high employment anxiety group was significantly higher than that of the low employment anxiety group (*t* = 9.35, *p* < 0.001, *d* = 1.12), which indicated that it was effective in inducing employment anxiety.

#### 2.5.2. Hypothesis Testing

As the results of the descriptive statistical analysis, the values of the intuitive level of career decision making under different conditions of anxiety and social distance are displayed in Table 1. The results of the above analysis provided a preliminary data basis for the subsequent hypothesis test.

Two variables, anxiety and social distance, were analyzed by two-way analysis of variance, and the results are shown in Table 1. The main effect of anxiety was significant (*F* (1.292) = 9.93, *p* = 0.001, η^2^_p_ = 0.04). Further tests found that the intuitive level score of the high-anxiety group was significantly higher than that of the low-anxiety group (high-anxiety group: *M* = 3.18, *SD* = 0.97; low-anxiety group: *M* = 2.79, *SD* = 1.00, *t* = 3.32, *p* = 0.001). Therefore, Hypothesis 1 was supported.

The main effect of social distance was significant (*F* (2.292) = 4.15, *p* = 0.013, η^2^_p_ = 0.03). Post-hoc analysis found that the intuitive level score of career decision making at a close social distance was significantly lower than that at a far social distance (close social distance: *M* = 2.79, *SD* = 1.17; far social distance: *M* = 3.15, *SD* = 1.02. *p* = 0.022), and that of medium distance relationship was significantly higher than that at a close social distance (medium: *M* = 3.10, *SD* = 0.90; close social distance: *M* = 2.79, *SD* = 1.17. *p* = 0.027). Meanwhile, there was no significant difference (*p* = 0.686) between those with medium distance relationships (*M* = 3.10, *SD* = 0.90) and those with a far social distance (*M* = 3.15, *SD* = 1.02). Therefore, Hypothesis 2 was supported.

The interaction between anxiety and social distance reached significance (*F* (2.292) = 2.97, *p* = 0.048, η^2^_p_ = 0.02), followed by a simple effect test, as shown in Table 2, indicating that in a close relationship, the score of intuitive career decision making with high anxiety (*M* = 3.12, *SD* = 1.11) was higher than that with low anxiety (*M* = 2.32, *SD* = 1.12) (*F* (1.292) = 13.62, *p* < 0.001, η^2^_p_ = 0.05). That is, helping persons in close relationships make career decisions in high-anxiety situations involves more intuitiveness than other decision making. In a medium distance relationship, the intuitive score with high anxiety (*M* = 3.21, *SD* = 0.84) was significantly lower (*F* (1.292) = 2.86, *p* = 0.091, η^2^_p_ = 0.01) than that with low anxiety (*M* = 2.92, *SD* = 0.82) at 0.1 level. However, there was no significant difference at a far social distance (*F* (1.292) = 0.12, *p* = 0.72). Therefore, Hypothesis 3 was supported.

The results of Study 1 provided strong causal evidence that anxiety and social distance affect the intuitiveness of career decision making and verified the theoretical hypothesis and the internal validity of the theoretical model. However, the external validity of this theoretical model needs to be further tested through a questionnaire survey of real job-seeking situations.

## 3. Study 2: Whole Model Questionnaire Survey

### 3.1. Participants

A total of 425 senior students who are looking for jobs were randomly selected to take part in the questionnaire survey from 5 universities in China, and the head teacher was asked to help send the students a link to an online survey website. A total of 386 valid questionnaires were obtained, with a recovery rate of 90.82%. Among them, there were 182 men and 204 women. The students came from 197 developed regions and 189 underdeveloped regions, and there were 218 undergraduates and 168 junior college graduates. The average age of the participants was 22.82 (*SD* = 1.31). The survey covered students in many disciplines, such as science and engineering, social science, and the humanities.

### 3.2. Measures

State Anxiety Scale. The same questionnaire was used in Study 1, and the Cronbach’s α coefficient in the study was 0.92.

The IOS Scale. The same questionnaire was used in Study 1. The lower the score, the farther the social distance is. The higher the score, the closer the social distance is.

Intuitive measure of career decision making. The scale used was adapted from the dimension of intuitive impulsivity in the General Decision-Making Style Scale developed by Scott and Bruce [53], which contains five questions, such as “in the process of choosing a career, I often make decisions on the spur of the moment”. Participants filled in the questions according to their actual situation. The Cronbach’s α coefficient in our sample was 0.87.

Control variables. Given that there are gender and regional differences in career decisions [3,55], we controlled for gender and region so that the effects of anxiety and social distance were separated from the effects of gender and region.

### 3.3. Statistical Analyses

Data were processed and analyzed mainly by the SPSS 23.0 Windows and Process 3.5 macro program [56]. Model 1 in the process macro program was used to test the moderating effect, and the bootstrap method was used to estimate the effect value of the 95% confidence interval to test the influence of anxiety on the intuitive strategy of career decision making with different social distances.

### 3.4. Results

In Table 3, the means, standard deviations, and correlations of the main variables are presented. Anxiety was positively correlated with intuitive career decision (*r* = 0.15, *p* < 0.05), and social distance scores were negatively correlated with the intuitive level of career decision (*r* = −0.13, *p* < 0.05), but there was no significant correlation between anxiety and social distance.

Regression analysis found that anxiety positively predicted the intuitive level of career decision making (β = 0.32, *SE* = 0.12, *t* = 2.52, *p* = 0.01), education negatively predicted the intuitive level of career decision making (β = −0.24, *SE* = 0.12, *t* = −2.08, *p* = 0.04), and gender and region did not significantly predict the intuitive level of career decision making, but some studies found that gender and region had an effect on career decision making [3,55]. Therefore, gender, education, and region were used as control variables. The anxiety score was then taken as a predictor variable, the social distance score was taken as a moderator variable, and the score of intuitive career decision making was chosen as a dependent variable. Then, the mean center of all variables and the bootstrap sample size were set to 5000. Finally, Model 1 was selected in the process macro program for testing. As shown in Table 4, the interaction of anxiety and social distance positively predicted the intuitive level of career decision making (effect = 0.34, *SE* = 0.16, 95% CI = [0.01, 0.67]).

To further explore the essence of the interaction effect of anxiety and social distance on the intuitive level of career decision making, the average plus or minus one standard deviation was set as the far social distance group and close social distance group, respectively, and a simple slope effect diagram was drawn, as shown in Figure 2. Under the three levels of far, medium, and close social distance, the fitting line between anxiety and the intuitive level of career decision making became steeper, indicating that the positive prediction of anxiety on the intuitive level of career decision making increases as social distance intimacy increases.

The Johnson–Neyman method was used for further analysis, and the results are shown in Figure 3. When the social distance intimacy value exceeded 1.8251, the confidence intervals of the effect of anxiety on the intuitive level of career decision making were above 0; that is, it did not include 0, indicating that anxiety had a significant positive prediction on the intuitive level of career decision making. In summary, when the social distance intimacy value reached 1.8251 or above, that is, when decision makers and job seekers were close in social distance, the relationship between anxiety and intuitive career decisions were enhanced, whereas it was not significant when they were far in social distance.

## 4. Discussion

Two statistical methods were used in this study. To further test and quantify the influence of anxiety on the intuitive level of career decision making at different social distances, grouping data and 2 × 3 analysis of variance were used first, followed by continuous data and bootstrap methods. The results of the two methods are consistent, indicating that the test results of our hypotheses in this study are robust.

### 4.1. Main Findings

The findings of this study show that when social distance is close, the relationship between anxiety and intuitiveness of career decision making is strengthened, and anxiety increases the intuitive level, whereas in the absence of anxiety, the intuitive level of career decision making is low. So, in the situation of anxiety, one cannot make the decision alone or ask close others to help with decision making; in the situation of no anxiety, one can make the decision alone or ask friends to help with career decision making. When social distance is far, although the influence of anxiety is small, the intuitiveness of career decisions is also large, so whether anxious or not you cannot ask strangers who do not know you to make decisions. This, to some extent, solves the problem of who should be called upon to make decisions in different situations.

This result demonstrates that social distance plays a moderating role between anxiety and intuitive career decision making, and that people who make career decisions for people who are close in social distance are more likely to be affected by anxiety. This result is consistent with past findings in cognitive neuroscience [57]. When individuals are making decisions for themselves, the associated brain regions responsible for emotional arousal control are more active than when making decisions for others [57]. Individuals are much more affected by their emotions in the former situation. This result further verifies the existence of emotional channels in decision making. In groups with close social distance, because of this close relationship, members are more concerned about each other than in groups with greater social distance, and they are more likely to feel empathy and have their anxiety triggered in an anxiety-inducing situation. Therefore, such individuals are more likely to experience emotional impulses and make intuitive decisions in career decision making. Under the influence of negative emotions, individuals are more inclined to local processing [58], which affects the overall judgment of individuals, and makes decisions more intuitive.

### 4.2. Theoretical Contributions

The theoretical contributions of this study are mainly reflected in the following three aspects. First, it enriches the relevant content of the cognitive–emotional dual-system model. In this study, social distance and anxiety are influencing factors in career decision making. They involve not only the concrete variables of cognition and emotion but also the theoretical application of the cognitive–emotional dual-system model. The research results further verify the existence of the cognitive–emotional dual-system and supplement the relevant content of the cognitive–emotional dual-system model [14]. When an individual is in a strong emotional state, the emotion system is dominant, and his or her decision making is mainly driven by emotion, which will lead to more intuitive decision making. However, when an individual is not in a weak emotional state, the cognitive system is dominant, and his or her decisions will be made through cognitive analysis, which will lead to more rational decision making [8]. Second, this study explores the mechanism through which anxiety influences career decision making, which is a powerful supplement to the research on how anxiety influences decision making, helps to deepen and enrich this research field, and expands the research field of the influence of anxiety. Anxiety affects decision making [24]. Few studies have applied anxiety to the career field, but this study can inspire more scholars to pay attention to the influence of anxiety on the career field. Third, it deepens and expands the theoretical research of career decision making. Career decision making is an important part of career psychology. Exploring the influence of anxiety and social distance on career decision making and the mechanism by which anxiety affects career decision making can enrich and expand upon career decision-making theory. This study can help us better understand and explain the influencing factors of career decision making in terms of both cognitive and emotional aspects and provide a theoretical basis for individuals to make more rational career decisions.

### 4.3. Practical Implications

This study has three practical implications. The first is that schools, employers, and individuals need to minimize job seeker employment anxiety to prevent graduates from making decisions that are too intuitive and impulsive. First, schools need to pay attention to the mental health education of college students and work to reduce their employment anxiety in time to slow down or reduce the negative impact of their anxiety on their career decision making. Second, in the recruitment process, the employer should try to create a relaxed and harmonious recruitment atmosphere, which can avoid the pressure of the interview and make individuals appear unnecessarily nervous and anxious, affecting their level of play and career decision-making mistakes, resulting in talent loss and increased recruitment costs. Finally, individuals need to recognize the impact of anxiety on career decision making. Individuals need to recognize the impact of anxiety on career decisions and seek timely help or hold off on making decisions when faced with employment anxiety situations. The second practical implication is to take the advice of close others when making career decisions, especially those close others who know you and understand your character, profession, and professional environment. Individuals make their own career decisions as much as possible with a full understanding of themselves and their career world if they are not anxious [11]. Individuals who do not understand themselves or their professional world should not make rash decisions and instead seek the advice of someone close to them. If everyone is in an anxious situation, it would be better for the individual to seek the help of professional career counselors who will assist the individual in making better decisions [59]. The third suggestion is for colleges and universities to carry out targeted career counseling to help students make effective decisions. Under the pressure of severe employment situations, when college students face employment and career decisions, employment anxiety is easily heightened. To avoid having anxiety influence their decision making, career counselors are needed to give targeted career guidance and constructive decision-making suggestions [60].

### 4.4. Limitations

There are some limitations in this study, and these provide directions for follow-up research. First, the research sample in this study mainly comprised college students, so the sample was highly homogenous. In the future, research samples should not be limited to college students but can expand the research object to include different job-hunting groups. Second, there are three strategies for career decision making, namely, rational, intuitive, and dependent strategies [11]. This study discussed the impact of anxiety and social distance only on intuitive and rational strategies in career decision making. Future research could continue to explore the impact of anxiety and social distance on dependent strategies in career decision making or study the effects of anxiety and social distance on a combination of the three strategies. Third, after reading the prompt to induce employment anxiety and responding to a career decision-making situation to measure individual career decision-making strategies, individuals’ experience might not be as strong as it would in a real situation, thus affecting the external validity of the findings. Thus, future studies could try to adopt a real job-hunting field experiment. Fourth, this study used cross-sectional data to analyze the impact of anxiety on the career decision making of people at different social distances. The static data make it difficult to ensure that these college students would make the same career decision in a real employment situation. In the future, longitudinal designs could be adopted to further explore the dynamic impact mechanism of anxiety and social distance on career decision making. Fifth, career decision making is a complex process. Between anxiety and career decision making, it is not only social distance that plays a moderating role but there may also be other mediators or moderators. In terms of social distance regulating career decision making, this decision making may also be affected by responsibility and empathy for others’ results, which needs to be further explored.

## 5. Conclusions

This study demonstrates that employment anxiety affects intuitive career decision making, and that the higher an individual’s employment anxiety, the more likely they are to make intuitive career decisions, and employment anxiety also affects decision makers to make decisions for people who are close to them. As a result, it is extremely important to take measures to prevent and alleviate employment anxiety. Moreover, individuals who make career decisions for people close to them have lower intuitive levels, so in situations where employment anxiety is low, they can make career decisions on their own or seek assistance from people close to them. However, in a situation of high employment anxiety, decision makers making career decisions for people close to them are vulnerable to employment anxiety and make intuitive decisions; thus, it is preferable to turn to others who are far socially distant to make decisions, particularly professional career counselors.

## Figures and Tables

**Figure 1 ijerph-20-01484-f001:**
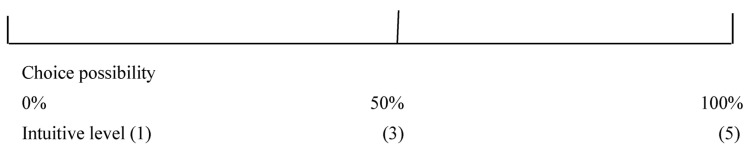
Operation diagram of the intuitive career decision-making.

**Figure 2 ijerph-20-01484-f002:**
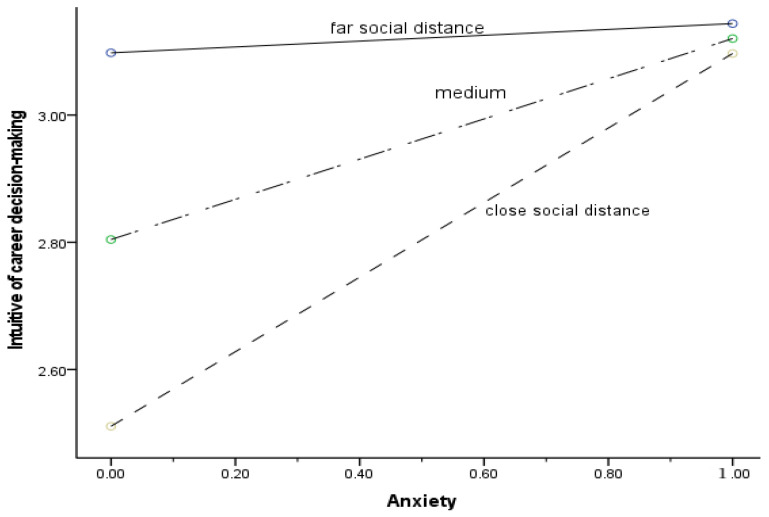
The moderating effect of social distance.

**Figure 3 ijerph-20-01484-f003:**
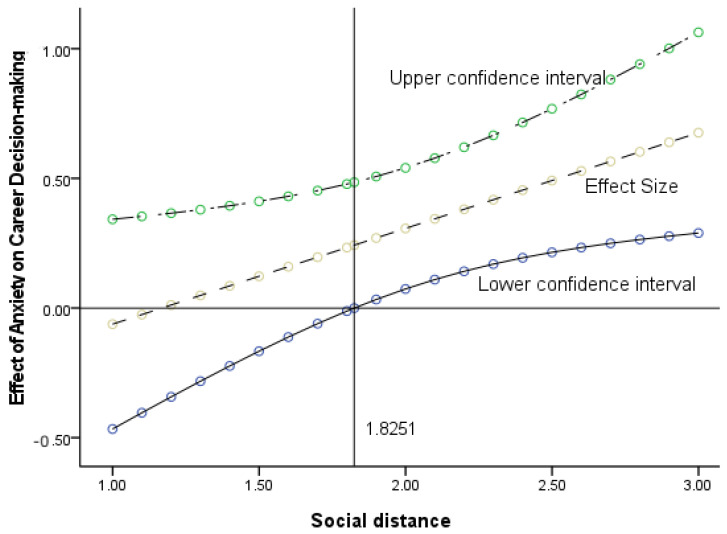
The relationship between anxiety and intuitive career decision making at different social distances.

**Table 1 ijerph-20-01484-t001:** Descriptive characteristics and ANOVA results.

Employment Anxiety	Social Distances	*N*	*M*	*SD*	*F*	*p*	η^2^_p_
High	Far	48	3.18	1.04			
	Medium	85	3.21	0.84			
	Close	49	3.12	1.11			
Low	Far	28	3.10	0.99			
	Medium	54	2.92	0.82			
	Close	34	2.32	1.12			
High		182	3.18	0.97	9.93	0.001	0.04
Low		116	2.79	1.00			
	Far	76	3.15	1.02	4.15	0.013	0.03
	Medium	139	3.10	0.90			
	Close	83	2.79	1.17			
Employment Anxiety × Social distances					2.97	0.048	0.02

**Table 2 ijerph-20-01484-t002:** Simple effect test.

Social Distance	Employment Anxiety		Mean Difference	Std. Error	*F*	*p*
**Far**	High	Low	0.08	0.23	0.12	0.724
Medium	High	Low	0.29	0.17	2.86 *	0.091
Close	High	Low	0.799 *	0.21	13.62	0.000

* indicates *p* < 0.1.

**Table 3 ijerph-20-01484-t003:** The means, standard deviations, and correlations.

Variable	*M*	*SD*	1	2	3	4	5
1. Gender	0.46	0.49					
2. Education	0.67	0.47	−0.07				
3. Region	0.47	0.50	0.11	0.15 *			
4. Anxiety	2.43	0.46	−0.05	0.02	0.03		
5. Social distances	2.02	0.73	−0.02	−0.11	0.08	−0.05	
6. Intuitive career decision	2.99	1.02	−0.07	−0.10 *	−0.11	0.15 *	−0.13 *

Gender, region and education were all set as dummy variables: 1 for men and 0 for women; 1 for those from developed regions and 0 for those from underdeveloped regions; and 0 for the junior college students and 1 for the undergraduate students. * indicates *p* < 0.05.

**Table 4 ijerph-20-01484-t004:** Test on the effect of anxiety and social distance.

	Intuitive Level of Career Decision Making		
	β	*SE*	95% CI
Gender	−0.14	0.11	[−0.37, 0.08]
Region	−0.18	0.12	[−0.43, 0.06]
Education	−0.32	0.12	[−0.57, −0.07]
Anxiety × Social distance	0.35	0.16	[0.03, 0.68]
*R* ^2^	0.10		

## Data Availability

The data are available from the corresponding author upon reasonable request (may require data use agreements to be developed).

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
