# Peer review of "Deciding Alone or with Others: Employment Anxiety and Social Distance Predict Intuitiveness in Career Decision Making"

_ijerph, 2023, doi:10.3390/ijerph20021484_

Round 1

Reviewer 1 Report

This paper have some contributions for the theoretical basis of promoting effective decision making from the views of employment anxiety and social distance. But there are the following questions which can be solved.

1. What is the novel of this paper? Please show it clearly.

2. Please the author read carefully all context, many language,words or paragraph must be revised. Some paragraph just only is one word ?

3. Why not the author report the result of low employment anxiety group in the section 2.5.1?

4. Suggest the author show the sample number and descriptive characteristics of basic students information in the different groups and add the table about the difference and significant of different group in the section 2.5.2 . Because this result is very important to the conclusion.

5. Why not the authors report and control the impact of the education, major and family situation on the level of intuitive career decision?

Suggest the authors to add the only the model with control variables and the moderating effect of social distance in the regression model in the table 3, which can check the result more precisely. 

Reviewer 2 Report

1. Please make the introduction more concise, focusing on the content you want to emphasize.

2. In order to assign subjects to the high employment anxiety group and the low employment anxiety group, information on the anxiety level is required in participants.

3. Are the research subjects 298 college students and 386 senior job-seeking students?

4. line 312 ' (t (296) = 9.35, p < 0.001, d = 1.12)' --> What does t(296) mean?

5. Please provide a rationale for the Anxiety Inducing Situation or explain why you created the situation that way.

6. Please check the agreement between the results and the statistical analysis.
